

# Fill dynamics and sample mixing in the AirCore

**Pieter P. Tans**

**Global Monitoring Laboratory, National Oceanic and Atmospheric Administration**

**325 Broadway, Boulder, CO 80305**

## Abstract

The AirCore is a long coiled tube that acts as a "tape recorder" of the composition of air as it is slowly filled or flushed. When launched by balloon with one end of the tube open and the other closed, the initial fill air flows out during ascent as the outside air pressure drops. During descent atmospheric air flows back in. We describe how we can associate the position of an air parcel in the tube with the altitude it came from by modeling the dynamics of the fill process. The conditions that need to be satisfied for the model to be accurate are derived. The extent of mixing of air parcels that enter at different times is calculated, so that we know how many independent samples are in the tube upon landing, and later when the AirCore is analyzed.

## 1 Introduction

When the Aircore is filling with atmospheric air coming in through the open end the newly sampled air pushes the air that is already in the tube deeper into the tube while compressing it. This mode of sampling is entirely passive, relying on the pressure continuing to increase as the altitude becomes lower during descent. The AirCore could also be flushed by a pump without any need for pressure changes of the outside air that is being sampled. I conceived the idea of AirCore in the late 1990s after we had found ~100 year old air, as indicated by the measured levels of $CO_2$ and $CH_4$, near the bottom of the firn layer at a depth of ~90 m at the South Pole (Battle, 1996). The air was very old despite the fact that there was still open contact with the present-day atmosphere. Over distances of tens of meters or more molecular diffusion is exceedingly slow! The root-mean-square (rms) molecular diffusion distance is $X_{rms}= (2Dt)^{0.5}$. D is diffusivity in air, for $CO_2$ it is 0.140 $cm^2$ $s^{-1}$ at 1 bar and 0 degree C, t is time in seconds. After one year the rms diffusion distance for $CO_2$ in air would be ~30 m which would be the scale of spreading if there is no macroscopic air motion at all. In addition, diffusivity deep in the firn is significantly slower than in open air because the air path from the bottom of the firn to the atmosphere has many detours going through the pores that are still open.

In collaboration with Jim Smith and Michael Hahn, we verified that there is very little mixing along the length of the tube by pushing slugs of air from two different reference air cylinders, alternating between high and low $CO_2$, through a long coiled tube. We also stored air for several



hours before analysis. It all looked good. Then we tried a balloon flight. In order to make the
payload lighter we switched from stainless steel to aluminum tubing, because of our excellent
experience with long term gas storage in high pressure aluminum cylinders. It did not work at all.
The easily bendable tube was made of a soft aluminum alloy, very different from the high
pressure cylinders. We found that the tube made $CO_2$ disappear very effectively. It was going to
take more effort to make it successful, and we did not have much time to devote to it. So the
project languished for several years until Anna Karion, Colm Sweeney, and Tim Newberger
were able to pick it up again. At the urging of Sandy MacDonald, who was director of NOAA's
Earth System Research Laboratory at the time, I applied for a patent in August 2006. He pointed
out that there are people trolling the scientific literature, conference proceedings, etc. to find
ideas that could be patented, so that we might find ourselves having to pay somebody else to use
our own idea. Instead, we wanted the AirCore to be freely useable (and improved) by everyone,
so that my patent (Tans, 2009) was intended to be a defensive action!
We realized that AirCore technology could become extremely useful for the validation of
satellite retrievals of column-averaged mole fractions of greenhouse gases. The measurements of
a gas sample captured by the AirCore are calibrated, but care has to be taken, as with all air
samples in containers, that no artefacts are introduced by the container or by gas handling
procedures. In contrast, remote sensing estimates of greenhouse gases can in principle never be
calibrated. Metrology, the science of measurement, defines what a calibration is. Using a
measurement standard, one presents the measurement method with a known value, under
controlled conditions, so that the measurement indication is related to a quantity value
(paraphrased from VIM3, JCGM 200:2008). In the case of greenhouse gases in the atmosphere
the conditions cannot be controlled. In addition, we realized that the regular deployment of
AirCores could be a cost effective way to monitor and study an evolving atmospheric circulation
as climate change progresses, as proposed by Fred Moore (2014).

## 2. The physical principle that makes the AirCore  work – molecular diffusion

Diffusive mixing over large distances is exceedingly slow, but there is another use of diffusion.
Flow inside the tube is laminar, which has maximum speed in the center and zero speed at the
wall. With velocities that differ from zero to some finite value, why does laminar flow **not**
"smear out" our tape recorder signal by mixing air parcels that came in at different times? Again,
molecular diffusion comes to the rescue. Using the square root relationship above, if the inner
radius of the tube is 0.3 cm, it takes a $CO_2$ molecule on average only 0.03 s (at 1 bar pressure) to
diffuse from the wall to the radius where the velocity equals the average velocity inside the tube.
Any molecule will be close to the wall, as well as near the center, of the tube many times per
second. Therefore the speed of all molecules in the long direction of the tube, averaged over a
few seconds, is very nearly the same. However, the AirCore idea does not work so well for
liquids. In water the molecular diffusivity is ~10,000 times lower than in air at 1 bar, so that the



smearing of a tape recorder signal could be very large. To compensate for such low diffusivity
both the diameter of the tube and the flow speed will have to be kept low, and there will also be
capillary effects.
The AirCore collects a continuous sample. Instead of valves, distance in the tube is used to keep
separated the air that has been sampled from different pressure altitudes. The number of
independent samples (the inverse of vertical resolution) in the tube decreases as the time between
collection and measurement becomes longer. The measurement, or "read-out" of the vertical
profile, is carried out by attaching an analytical instrument to one end of the tube and a cylinder
with air of well-known composition to the other end. The latter pushes the sampled air slowly
through the analyzer. The procedure, as well as various tests of mixing, has been described by
Karion (2010).

**3. Dynamics of the fill process**
How do we accurately associate position in the tube with the geometric altitude or pressure
altitude that the sample at that position came from? It is the first question we address in this
paper. The filling does not occur uniformly as a function of pressure altitude. The second
question is how far the mixing of adjacent air parcels extends as a result of molecular diffusion,
and secondarily as a result of the flow itself. I wrote the first version of the algorithm to make the
association of altitude with position in 2005, called rocketfall.pro, coded in Interactive Data
Language (IDL). Undergraduate students in the engineering department at the University of
Colorado were getting ready to put an AirCore on a NASA rocket, and we were worried about
there not being enough time to passively collect air from the stratosphere as the rocket was
falling at supersonic speeds. There have been several successive versions of the algorithm since
then. The version of July 2021 is described here.
We use a fluid dynamics model and flight data, namely the pressure and temperature of outside
air and the temperature of the tube as input data. The starting point is Poiseuille's equation for
steady state laminar flow in a tube with circular cross section:
$$Q_m = \frac{-\rho\,\pi\,r^4}{8\,\eta}\,\frac{dP}{dz}\,, \quad \text{or} \quad Q_n = \frac{-\rho_n\,\pi\,r^4}{8\,\eta}\,\frac{dP}{dz} \qquad\qquad \textit{Eq. 1}$$

in which $Q_m$ is mass flow (kg s$^{-1}$), $Q_n = Q_m/M$ is amount flow (mol s$^{-1}$) with M molecular weight
of dry air (0.02896 kg mol$^{-1}$), $\rho$ is gas density (kg m$^{-3}$), $\rho_n$ is amount density ($\rho/M$ in mol m$^{-3}$), $\eta$
is viscosity (kg m$^{-1}$ s$^{-1}$), r is tube radius (m), P is pressure in Pascal (kg m$^{-1}$ s$^{-2}$), and z is
distance along the tube (m). Pressure is given by the ideal gas law as P = (n/V) RT, with n/V = $\rho_n$
the number density in mol m$^{-3}$, T is temperature in degrees Kelvin (K), and R the universal gas
constant, 8.3144 J mol$^{-1}$ K$^{-1}$. The flow velocity is parabolic as a function of radius, zero at the
wall, and maximum in the center where the speed is twice the average speed.





The viscosity ($\eta$) depends on temperature, but it is very nearly independent of pressure in our
range of interest. The latter is of primary importance to the fill process. A simple approximate
molecular expression for viscosity is $\eta \cong (1/3) \rho \, \mathbf{c} \, \lambda$, in which $\mathbf{c}$ is the average molecular speed
and $\lambda$ is the mean free path between collisions which is inversely proportional to $\rho$ (Jeans, 1952).
Since the volume flow (m$^3$ s$^{-1}$) is $Q_v = Q_m/\rho$, Eq. 1 states that the volume flow depends on
viscosity, but not on gas density. It takes the same amount of force (pressure difference) to push
the same volume flow irrespective of the density of air in that volume. During steady flow
through any tube the flow needs to speed up at the low pressure end to conserve mass so that the
pressure gradient always steepens at the low pressure end.
The z-coordinate is for position along the length of the tube. The pressure change at any point in
a small section of the tube with length dz can be due to temperature change or to more amount
flow coming in from z than leaving from z+dz. The latter term is
$$\frac{d\rho_n}{dt} = -\frac{1}{\pi r^2}\frac{dQ_n}{dz} \, , \quad \text{so that}$$

$$\frac{dP}{dt} = \rho_n R \frac{dT}{dt} + RT \frac{d\rho_n}{dt} = \frac{P}{T}\frac{dT}{dt} - \frac{RT}{\pi r^2}\frac{dQ_n}{dz} \qquad \text{Eq. 2}$$

If we assume that the tube is round (not elliptical for example) the amount flow $Q_n$ is given by
Poiseuille's equation, and Eq. (2) can be represented numerically in a very efficient manner. In
that case the flow is in effect solved as a succession of steady state flows that evolve slowly in
time and along the length of the tube. In the rest of this section we will discuss a number of
assumptions we are making for our "succession of steady state flows" approximation to Eq. (2)
to be satisfactory.
The first one is that inertial effects, i.e. accelerations, die out very rapidly. Suppose we suddenly
set the pressure gradient that is driving the flow to zero. What is the time scale for the flow to die
down? We can estimate the time it takes for the flow to adjust by using Eq. 1. The average speed
of the flow is $\mathbf{v}_{avg} = Q_v/(\pi r^2) = (r^2/8\eta)(\Delta P/\Delta z)$. The momentum of the flow in length $\Delta z$ is $\mathbf{v}_{avg} \rho$
$\pi r^2 \Delta z$ which equals $Q_m \Delta z$ (neglecting the sign). The rate of change of momentum is given by
the frictional force which is equal and opposite to the pressure force that was driving the flow in
Eq. 1. The adjustment time scale of the flow is momentum divided by the frictional force,
$$\tau = \frac{Q_m \, \Delta z}{\Delta P \, \pi r^2} = \frac{\rho \, r^2}{8 \, \eta} \qquad \text{Eq. 3}$$

For a tube with a radius of 3 mm and $\rho$ corresponding to 1 bar and 285 K, $\tau \cong 0.07$ s. At an
altitude where the density is 10 times lower (~18 km), $\tau \cong 0.007$ s. Recently NOAA GML has
been flying AirCores with $r \cong 1.46$ mm, for which the adjustment time at 1 bar and 285 K is $\tau \cong$
0.017 s. A succession of steady state flows is indeed a very close approximation.



Next we assume that the temperature of the gas is the same as that of the wall. How rapidly does
the temperature of the gas equilibrate with the wall of the tube? The heat capacity of a volume of
air is $c_p \rho_n \cong (7/2)$ R * P/RT in which $c_p$ is the molar heat capacity at constant pressure and $\rho_n$ is
the number density (mol m$^{-3}$) of the gas, so that $c_p \rho_n$ has units of J m$^{-3}$ K$^{-1}$. The heat
conductivity of gas is $\kappa \cong (1/3) c_v \rho_n \mathbf{c} \lambda$ (Jeans, 1952) in which $c_v$ is the molar heat capacity at
constant volume, $\mathbf{c}$ is the average speed of individual molecules and $\lambda$ the mean free path. It has
units of (J/s) m$^{-2}$ (K/m)$^{-1}$, the heat flow per area per temperature gradient. As in the previous
paragraph we divide the heat energy change corresponding to $\Delta T$ in a volume of gas residing in a
length $\Delta z$ by the heat flow from the wall assuming the temperature gradient is close to
($\Delta T/(0.5r)$). That gives
$$\tau = \frac{c_p \rho_n \, \pi r^2 \Delta z \, \Delta T}{(1/3) \, c_v \rho_n \, \mathbf{c} \, \lambda \, 2\pi \, r \, \Delta z \, \Delta T/(0.5r)} = \frac{c_p}{c_v} \frac{3 \, r^2}{4 \, \mathbf{c} \, \lambda} \qquad \text{Eq. 4}$$

which has units of seconds. For r = 3 mm and $\lambda$ corresponding to 1 bar, and 285 K, the
adjustment time is $\tau \cong 0.31$ s, and shorter at lower pressures. For r = 1.46 mm $\tau \cong 0.07$ s.
Is the flow always laminar as Eq. 1 assumes? If Reynolds number, Re = $(\rho \mathbf{v}_{avg} d)/\eta$, in which d is
the diameter of the tube, stays below 1000, the flow will remain laminar. Re is estimated from
the calculated velocities, $\rho/\eta$, and tube dimensions for every flight. It is highest just before
landing when it typically has a value of ~15.
The tube is wound up in a coil with typical diameter 20 to 30 cm. As the flow goes around the
coil there will be a centrifugal force away from the center of the coil. The centrifugal force is
greatest where the flow has the maximum velocity, 2 $\mathbf{v}_{avg}$, very near the center of the tube. This
sets up a secondary flow in the plane perpendicular to the main flow, outward in the center of the
tube and back along the walls. The location of maximum velocity is also pushed outward a bit.
This increases flow resistance leading to slightly lower $Q_m$ for the same pressure gradient in the
dimension z along the length of the tube. However, there are other subtle effects with the
opposite sign that could facilitate the flow a little (Berg, 2004). Correction factors to flow in a
straight tube have been calculated using Dean's number, De = Re $(r/R)^{0.5}$, in which Re is
Reynolds number and R is the coil radius. NOAA GML has flown AirCores with r/R from 1/50
to 1/70. Thus De is always smaller than 15 $(0.02)^{0.5} \cong 2$ during a flight. Berg et al. (2004)
present data to estimate that the relative flow correction is smaller than +1 10$^{-5}$ for our
parameters. If we were to wind our coil much tighter, say with r/R of 1/20, then the maximum
relative flow correction during a flight would be +2 10$^{-4}$ for the same Reynolds number.
Therefore we can neglect the corrections for the tube coil curvature.
If the tube is elliptical instead of circular, there is a good approximation for the change in flow
resistance. Following Lekner (2019), Eq.1 can be written for volume flow as $(\eta \, Q_v) / (dP/dz) =$
$\pi \, r^4 / 8$, neglecting the sign. Note that $\pi \, r^4 / 8$ equals $A^3 / (2 \, P^2)$ for a circular cross section, with
A the cross sectional area, and P the perimeter of the tube. Lekner shows that $A^3 / (2 \, P^2)$ applies





quite generally for many cross sectional shapes. So if the tube is somewhat squashed into an
ellipse with major axis 1.05 times the original radius, and a minor axis slightly smaller (in order
to keep the perimeter the same) than 0.95 times radius, the term $A^3 / (2 P^2)$ has become ~1%
smaller. This is not a major effect.
We assumed the ideal gas law. Non-ideality is often described by the virial expansion relating
pressure and density, $PV/nRT = 1 + B(n/V) + C(n/V)^2 + \ldots$  Note that n/V is called $\rho_n$ above.
Taking only the second (and largest) virial coefficient B ($m^3$/mol) into account we can
approximate the number density $\rho_n$ as (P/RT)(1-BP/RT). The relative change of number density
is thus BP/RT which has dimension one. At 300 K and 1 bar, B is -7.3 $10^{-6}$ $m^3$/mol
(Sevast'yanov, 1986) which leads to a relative density increase of 2.9 $10^{-4}$. B increases to -18.9
$10^{-6}$ and -37.8 $10^{-6}$ $m^3$/mol at 250 K and 200 K respectively, but at the higher altitudes the
density is lower so that the largest non-ideality effect occurs near the ground. Therefore the
fractional density increase relative to ideal gas during a flight remains well below 0.001
When the mean free path increases at lower pressures there could be "wall-slip", non-zero
velocity at the wall which can be modeled as an effective decrease in viscosity increasing the
volume flow. Berg (2005) gives an approximate expression for the factor by which the flow
increases, $1 + 4 K_{slip} Kn$, where $K_{slip}$ is a number close to 1 which depends on intermolecular
forces, and Kn is the Knudsen number, $\lambda/d$, with d being the internal diameter of the tube. At
high altitude, say 10 millibar (mb), $\lambda \sim 7 \cdot 10^{-4}$ cm, so that Kn ~ 0.001 for d = 0.6 cm. For d = 0.3
cm the flow would be increased by a factor 1.009 at 10 mb.
When Kn becomes larger than ~0.01 a transition region of pressure is entered in which the flow
changes gradually from bulk flow of gases, laminar in our case, to molecular flow (O'Hanlon,
1980). In the latter flow regime the gas sample enters the tube as individual molecules, and gases
with higher molecular speed (lower mass) enter the tube more rapidly, so that the air sample may
not represent the composition of outside air, whereas in bulk flow an overwhelming fraction of
all molecules are equally swept along.  As an example, for an AirCore with opening diameter of
0.3 cm this flow transition starts at a pressure altitude of ~2 mb. Therefore, approximately 43 km
might be the highest altitude that can be sampled with this diameter opening without first
quantitatively investigating molecular flow effects, although this limit depends also on the
sampling accuracy we require.
The above expressions for viscosity, $\eta \cong (1/3) \rho \mathbf{c} \lambda$, and heat conductivity, $\kappa \cong (1/3) c_v \rho_n \mathbf{c} \lambda$,
and similar for diffusivity, $D \cong (1/3) \mathbf{c} \lambda$ are approximate. More precise forms of these equations
vary depending on the treatment of intermolecular forces. Instead, we use a curve fit to empirical
data for viscosity in dry air as a function of temperature, as presented by Kadoya (1985). The
empirical data show, as expected, that there is no dependence on pressure in our range of
interest.





For diffusivity of trace gases in air as a function of temperature and pressure we use the
empirical equation presented by Massman (1999), $D(T,P) = D_0 (P_0/P) (T/T_0)^{1.81}$. $D_0$ is the
diffusivity, different for each trace gas in air, at 1 atmosphere air pressure ($P_0$) and 0 degrees
centigrade ($T_0$). This will be used when we calculate mixing of air samples entering the AirCore
sequentially. Mixing is caused both by molecular diffusion ($X_{rms} = (2Dt)^{0.5}$, see above) and by
the quadratic velocity profile of laminar flow, with zero speed at the wall and maximum speed in
the center. The latter is called Taylor diffusion (Karion, 2010), and is given by a diffusivity
constant $D_T = \mathbf{v}_{avg}^2 \, r^2 /(48 \, D)$ which has the same dimensions as D, $m^2 s^{-1}$.

**4. Calculated in- and outflow results for some flights**
In Figures 1- 4 the flight is shown of a small diameter (1/8 inch, internal diameter 2.92 mm)
AirCore (GMD008), with 93 m length and internal volume 619 cc, near Trainou, France (48.0
°N, 2.1 °E) on 20 June 2019. The ascent velocity of the helium balloon is nearly constant, while
the rate of mass outflow decreases steadily as a function of time as the pressure outside and
inside the AirCore drops. The descent velocity with parachute accelerates nearly linearly in the

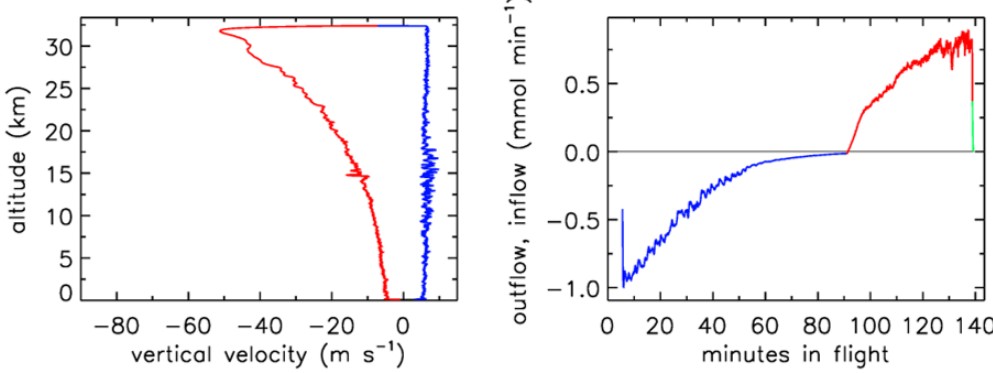


*Figure 1. Descent velocity (negative) and rate of fill air outflow followed by air sample inflow*
*during flight of GMD008.  Blue, ascent; red, descent; green, on the ground*
first 10 seconds to about 50 m s-1 as the pressure at high altitudes is too low for air friction to
slow it down enough. The initial descent can be a chaotic tumble until the parachute gets a
"grip". Outflow and inflow in the tube are calculated with the fill dynamics program described
below in section 8.

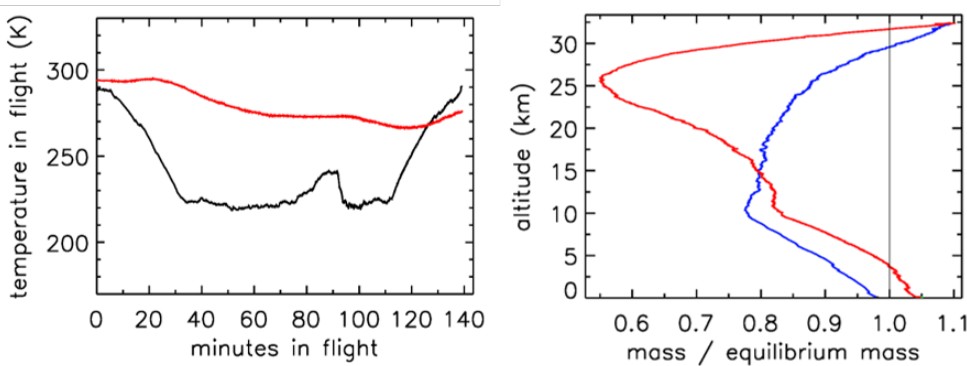


*Figure 2. Flight of GMD008. Left panel: Temperatures in degrees Kelvin. AirCore tube, red;*
*outside air, black. Right panel: Blue, ascent; red, descent.*


In Figure 2 the outside air temperature first cools while in the troposphere, then becomes nearly
constant in the tropopause, and starts increasing again higher into the stratosphere. GMD008 was
well insulated but still partially followed the outside temperatures with a delay. In the right panel
the total amount of air in the tube is plotted relative to how much it would be if it had the same
pressure and temperature everywhere in the tube as the outside air. Vertical line: the ratio equals
1 if they were the same. During ascent in the troposphere (up to about 10 km) the air in the tube
is warmer, and thus less dense, than outside air. In the tropopause the tube continues to cool so
that the "deficit" becomes smaller, but at higher altitudes, around ~25 km the amount by which
the pressure in the tube is higher than outside becomes substantial relative to the low outside
pressure – as a result the ratio at ~34 km altitude becomes a bit larger than 1. Then, during
descent the outside pressure increases rapidly and the inflow cannot keep up because the
viscosity of air at low pressure is the same as at 1 bar (see section 3). Back in the troposphere the
tube warms up, but much more slowly than outside air. When the tube hits the ground, it is
colder than ambient air temperature so that the ratio is greater than 1.
In Figure 3 the fill rate is plotted (mol per hPa of ambient pressure gain) divided by the final fill
(moles of air) at valve closure. At sea level the final pressure is close to 1013 hPa, so that the

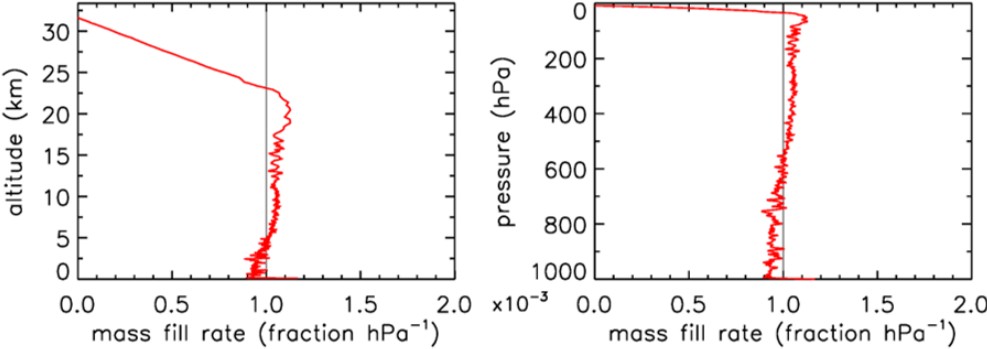






*Figure 3. Flight of GMD008. The vertical line at 1.0 10$^{-3}$ is approximately the expected rate of*
*sample inflow.*
the average fraction of final fill per hPa will be approximately 0.001. The uptick upon landing is
the result of a bit of air still entering the tube initially while ambient pressure stops changing,
neglecting high frequency noise. If the valve is not closed quickly this will reverse because as the
tube warms up on the ground, the last air that came in will be expelled. At high altitudes it takes
time for the fill to start because ambient pressure needs to build up enough to force the air in.
The highest altitude was 32.4 km, at 7.7 hPa ambient pressure. The fill starts at 31.6 km and
pressure 8.5 hPa, slowly at first, and gradually becomes faster. To compare the start of fill
between AirCore designs with different diameters and valves, we could take the point at which
the fill rate is 0.5 10$^{-3}$. In this case the "half-fillrate point" is at 27.3 km and ambient pressure of
17.3 hPa. We will see that the fill starts much faster with larger diameters. Figure 4 shows detail

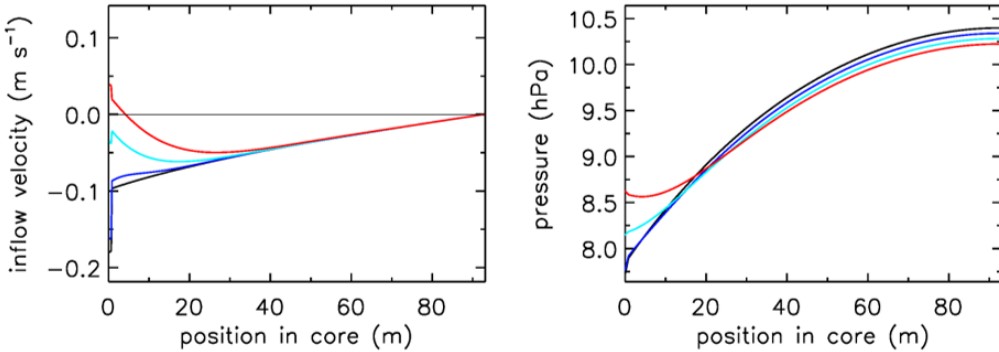


*Figure 4. The turnaround at high altitude. Inflow velocity and pressure inside the AirCore from*
*the moment the ascent stops and descent begins. Black, 0 seconds after start of descent; dark*
*blue, 7 s after start; light blue, 14 s; red, 21 s.*
of flow and pressure inside the tube for the flight on 20 June 2019 at the start of descent. Initially
the inflow velocity is negative. It is outflow, zero at the closed end and increasing toward the
open end. The velocity has to jump up inside the flow restrictions of the valves and possibly the
dryer at the entrance of the AirCore, adjacent to position at 0 m. After 14 seconds into the
descent (light blue curve) the outflow has weakened considerably and the pressure gradient near
the open end is much smaller. Inflow starts after 19 s, very slowly at first, while at the same time
the flow in most of the tube is still negative (outflow toward the open end), consistent with the
pressure gradients.
Let us look now at an AirCore with larger diameters (Figure 5). This one had 26 m of 1/4 inch
(internal diameter 5.84 mm internal diameter) tubing at the open end and 37.6 m of 1/8 inch
(2.67 mm internal diameter) tubing at the closed end, with a total internal volume of 980 cc. In
front of the open end was a valve, the dryer (large magnesium perchlorate particles), and then
another valve connecting to the AirCore tube. It was flown in Oklahoma, U.S. (37.2 N, 97.8 W),
on 23 July 2013. While the AirCore used near Trainou, France, experienced a temperature range

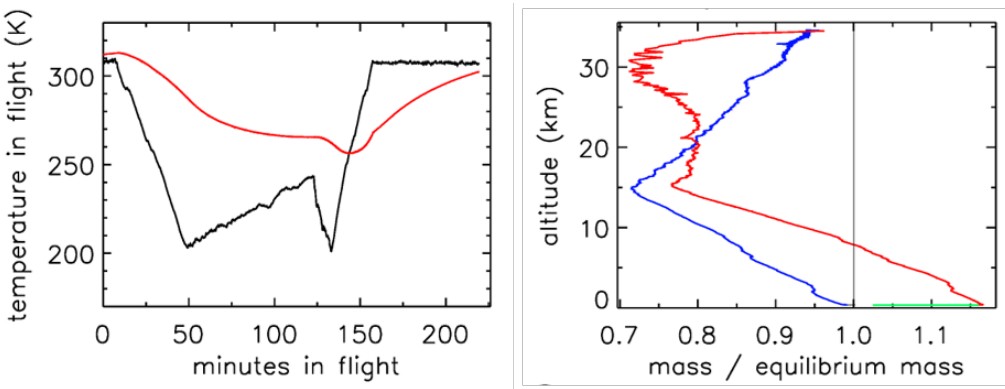


*Figure 5. Flight of AC01 in Oklahoma. Color scheme in right panel: Blue, ascent; red, descent;*
*green, lying on the ground.*
of 15 K, the less well insulated AC01 in Oklahoma saw a range of 57 K. At the moment of
landing the average temperature of the tube was ~40 K cooler than ambient. Fig. 5 shows the
flight data until the moment of valve closure. The valve remained open for 62 minutes after
landing, so that the lowest portion of the atmospheric sample, between pressure altitudes of 844
and 967 mb (1565 to 352 m), was expelled as the AirCore warmed up. The descent started at
34.6 km altitude (4.6 mb). The lowest relative mass deficit (~27%) was reached around 30 km, in
contrast to the Trainou flight with 50% at 27 km altitude respectively. The half-fillrate point of

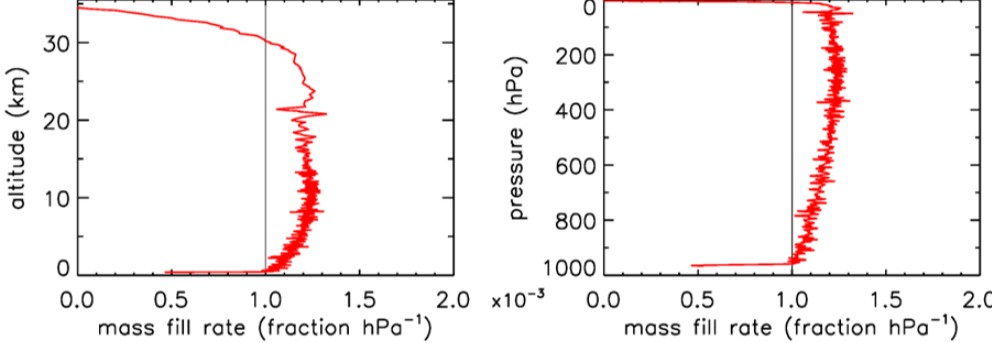


*Figure 6. Flight of AC01 in Oklahoma. Compare with Figure 3.*
$0.5 \cdot 10^{-3}$ per hPa is reached at 33.2 km altitude and 6.2 mb of ambient pressure, a sampling
altitude gain of almost 6 km compared to the Trainou flight. If the total amount of fill air that
remained in the tube is carefully measured that would give an independent determination of the
half-fillrate point estimated here. The fill rate below ~8 km falls off noticeably as the warming



rate of the tube speeds up. The negative mass fill rate while on the ground cannot be portrayed in
Fig. 6 because ambient pressure remains constant. This AirCore design contains a larger fraction
of stratospheric air than GML008, mostly because of the wider diameter, but also because it was
allowed to cool more.

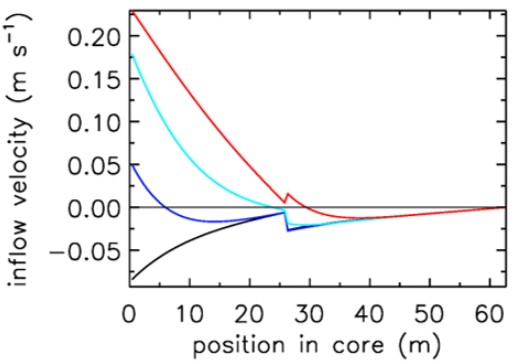
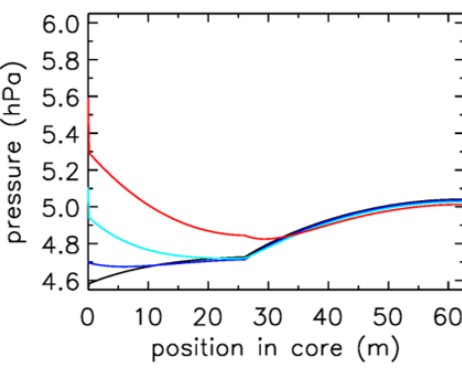


*Figure 7. Flight of AC01 in Oklahoma, showing inflow velocity Compare with Figure 4. Black, 0*
*seconds after start of descent; dark blue, 2 s after start; light blue, 5 s; red, 10 s. Note the much*
*smaller delay than in Fig.4. The flow velocity inside the valves and dryer has not been plotted*
*here, but the pressure drop across them has.*
If one wants to sample still higher into the stratosphere the diameter of the first 10 to 20 m at the
open end needs to be widened further. All of this is consistent with Fig. 7, where we also see that
at the start of the descent the outflow velocity inside the tube drops by a factor of ~4 when,
moving from the back to the open end, at 26 m the tube diameter becomes wider by a factor of 2.
This applies of course also to the inflow as shown by the red curve. At the same point the
pressure gradient becomes less steep by the same factor of 4. The fill starts at ambient pressure
of 4.7 mb. We also note that in this case the pressure drop inside the two valves and the dryer is a
large part of the overall pressure drop across the entire tube, especially when the tube diameter is
larger.
In these calculations we have experimented with another strategy to fill the AirCore. One could
launch it with both valves open, but the one in the back is closed as soon as the descent starts.
That would decrease the amount of fill air that remains in the back. However, the difference from
having the back valve closed during the entire flight is minuscule.

**5. Valves**
So far the treatment of valves and the dryer has been missing from this description. As a first
approximation we could treat the valves as short pieces of tubing with reasonably "average"
internal diameter and length such that their internal volume is correct. This does not provide
enough flow resistance, when we compare it to differential pressure measurements made during
some flights between the closed end of the AirCore and the outside ambient air (Figure 8).

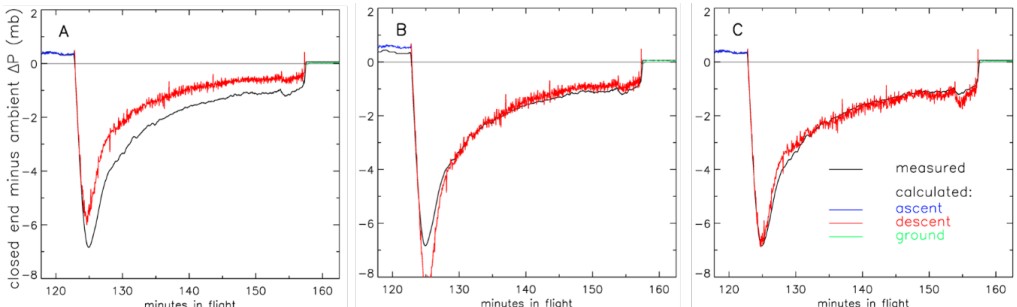


*Figure 8. Pressure difference (ΔP) between closed end of tube and outside air during the*
*descent portion of flight of AC01 in Oklahoma as a function of elapsed time in flight. Black:*
*measured pressure difference (mb, or hPa). Red: calculated ΔP with three different treatments of*
*the valves.* A, *fixed internal diameter and length;* B, *same as in* A, *but optimized;* C, *using*
*optimized* Cv *and* $X_{TPR}$ *(see text) values.*
In panel A we calculate that during the descent the air enters the tube too easily, so that the
altitudes assigned to the air sample in the stratosphere would be biased high. We could decrease
the chosen internal average diameter of the valves (panel B), optimized so that the difference
between calculated and measured ΔP during the entire descent, from minute123 to157, is
minimized. However, it is clear that this effective or apparent internal diameter needs to change
during the flight. Using Cv values and a description of choked flow is clearly better. In panel C
we have chosen the Cv and $X_{TPR}$ values such that the average difference from minute 123 to 157
is zero and the standard deviation of differences is minimized. This implicitly includes any
effects caused by the dryer in between the two valves.
The flow inside a valve can be complicated, with sharp corners, turbulence, sudden acceleration
through a flow restriction with its associated heating and cooling of the gas, etc. The industry has
introduced flow coefficients (Cv in the U.S., and Kv elsewhere) as an empirical approach to flow
calculations, as in the Swagelok brochure (2020). The expressions for air, slightly generalized
from Swagelok, for gas flow are as follows. For low pressure drop flow, we have
$$Q_n = 6950 \ C_v \ P_1 \left(1 - \frac{X}{3 \ X_{TPR}}\right) \sqrt{\frac{X}{T_1}}$$  (Eq. 5a), where $Q_n$ is in liters per minute at standard
conditions of 1 bar and 0°C, $P_1$ and $T_1$ are pressure (bar) and temperature (Kelvin) upstream of
the valve, ΔP is the pressure drop across the valve, X is the pressure drop ratio $\Delta P/P_1$, and $X_{TPR}$
is the terminal pressure drop ratio between (0 and 1) above which we have choked flow. Under
choked flow conditions the flow is fully independent of P and T downstream of the valve. It is
also important to know that the flow coefficient $C_v$ is not a pure number, but has physical
quantities and units embedded in it.





For a high pressure drop ($X > X_{TPR}$), we have
$$Q_n = 6950\, C_v\, P_1\, \frac{2}{3}\, \sqrt{\frac{X_{TPR}}{T_1}}$$    (Eq. 5b), which is obtained from the previous expression by
substituting $X_{TPR}$ (a constant) for X. In these expressions we prefer to express the flow, instead
of in standard L/min as in the Swagelok brochure, as 0.04403 mol/min. This is the same, when
using the molecular weight of dry air (28.97 g/mol), as a mass flow of 1.276 g/min.
In Fig. 8C we optimized both $C_v$ and $X_{TPR}$ to get the best match for the calculated pressure
difference across the AirCore with the observed history during the descent. The value of $X_{TPR}$
depends on valve design, and may not be the same when flow goes in the opposite direction.
Many valves have an arrow for flow direction printed on them. For most AirCore flights
differential pressure measurements have not been recorded. But the valves (and also driers) could
be tested with a standard procedure (see Figure 9 as one example). Alternatively, or as a
complementary check, a micro-spiking method during filling could be used (Wagenhäuser,
368    2021).

Figure 9 shows a potential test procedure for determining $C_v$ and $X_{TPR}$ values. The figure is
drawn using the two expressions for $Q_n$ above, for low flow and choked flow. Starting from a
uniform pressure of 1 bar, the pressure at the downstream side is lowered in 10 mb steps, at 2 s
intervals. In this example $C_v = 0.01$ and $X_{TPR} = 0.5$, so that the transition to choked flow occurs
at a pressure drop of 0.5 bar (panel A, upward arrow at 100 s). When the pressure at 10 m
approaches zero, the flow speed is high, causing a significant pressure drop between 5 and 10 m.

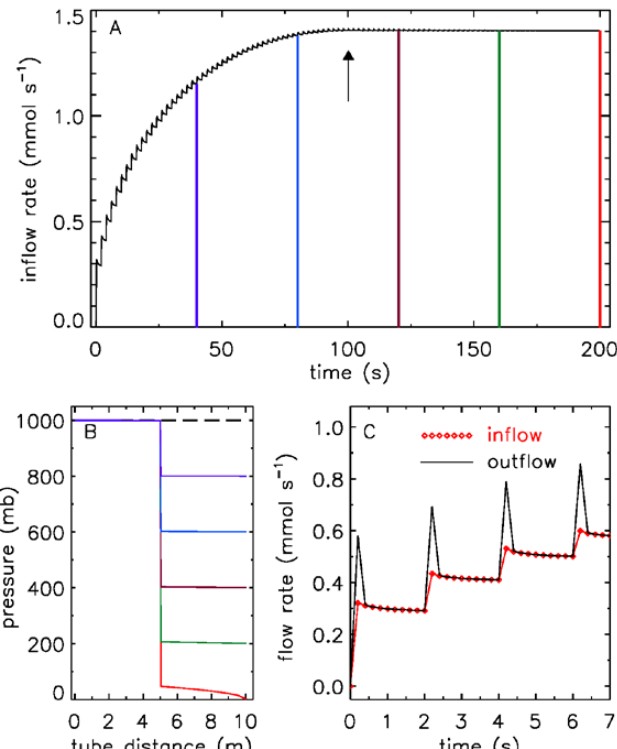


*Figure 9. A potential test procedure to determine Cv and $X_{TPR}$ values for valves. In this example*
*there is 5 m of ¼" tubing on each side. Outflow at the 10 m point (black curve) is shown in*
*panels A and C. There is a flow pulse at every step because the downstream 5 m section empties*
*quickly. The time resolution is 0.2 s. Inflow at 0 m is shown as red diamond symbols in panel C.*
*Panel B, pressure in the tube from time 0 (dashed line) at 40 s intervals, corresponding to the*
*colors in panel A.*



## 6. Mixing inside the tube


The fill dynamics calculation has produced time series of air density, pressure and temperature,
and flow velocity everywhere in the tube as a function of time, from the start of the fill process,
which begins a varying amount of time after the AirCore has started its descent, to the time of
valve closure. We divide the final amount of air in the tube at closure into 400-500 equal mass
packets. Starting from 400 we increase the number, which shrinks the size of each packet, until
the remaining fill air in the back of the tube comprises an exact integer number of packets. For
each mass packet, after it has entered the tube we follow it through the tube, as it is pushed
toward the back while being compressed by packets entering later. The time steps are defined by
when a new packet has fully entered, and they are longer at the start of the fill. The molecular





394 diffusivity D and the Taylor diffusivity $D_T$ are different at each step. However, the amount of
395 spreading of a packet calculated at each time step "k" is decreased as the increasing pressure
396 compresses the packet further. So the contribution of each step to the final spreading at valve
397 closure is calculated by dividing the density during that time step by the final density in the tube.
398 We are thus accumulating the "2Dt" term of $X_{rms} = (2Dt)^{0.5}$, with Taylor diffusion added:

399
$$X_{rms} = \sqrt{2 \; \Sigma_k\left(D_k + D_{T,k}\right) \frac{\rho_k}{\rho_{final}} \; t_k} \qquad\qquad Eq.\,6$$

400 For an AirCore with (almost) uniform diameter we get mixing as in Fig. 10 A. Close to the open
401 end at position 0 m, there is very little mixing because the time to mix was short. Near the closed
402 end at 93 m the spread of mixing deviates from what see in the first approximately 2/3 of the
403 tube because the fill started slowly, giving extra mixing time for the high altitude samples that
404 were later pushed to the back.

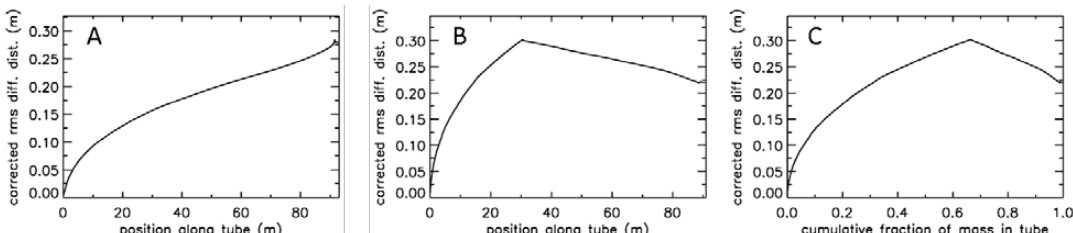

405

406 *Figure 10. Root-mean-square diffusive mixing when the valve at position 0 is closed. Panel A,*
407 *Flight of GMD008 in Trainou. Panel B, the same flight data, but used to calculate the filling of a*
408 *different AirCore, with 30.9 m of 1/4" tubing at the open end, and 60.1 m of 1/8" at the closed*
409 *end. Panel C, same as B, but plotted as cumulative fraction of total mass, from 0 to1.*

410 For an AirCore with two sections of different diameter we see an interesting effect (Fig.10 B).
411 The air that comes in at high altitudes and ends up in the back of the tube, has to go through
412 the1/4" section first. When a packet enters the 1/8" section, its spread becomes approximately
413 four times larger, while its 2Dt accumulation term stays the same. Approximately, because the
414 inner diameters (ID) matters, not the outer (OD). To correct for the jump we add another factor
415 to Eq. 6, and we will call this corrected rms diffusion distance:

416
$$X_{rms} = \sqrt{2 \; \Sigma_k\left(D_k + D_{T,k}\right) \frac{\rho_k}{\rho_{final}} \; \frac{(dvol/dx)_k}{(dvol/dx)_{ref}} \; t_k} \qquad\qquad Eq.\,7$$

417 In Eq. 7 dvol/dx is the increment in volume per increment in length of the tube, while $(dvol/dx)_{ref}$
418 is the total volume divided by the total length, both in units of $m^2$. This prevents a jump at the 30
419 m position, but more importantly, what matters for mixing is the spread relative to total mass in
420 the tube, not whether it is in the 1/4 or 1/8" section. From now on we call this configuration "1/4
421 -1/8". Fig. 10 B shows that air closer to the back has been in the 1/4" section for a shorter time,





and thus experienced less mixing relative to mass. When plotting mixing not as a function of
position, but as a function of cumulative mass in the tube, Fig 10 C also shows that the 1/8"
section contains approximately 1/3 of the total air sample.

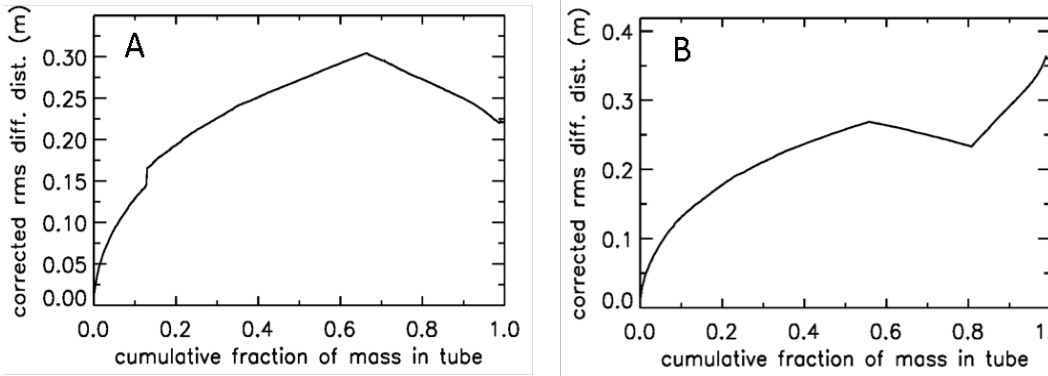

*Figure 11. Two additional cases of mixing upon valve closure. Panel A, same AirCore 1/4 -1/8,*
*but the flight data have been changed. Panel B, same flight data as in Fig. 10A, but the AirCore*
*configuration is 1/4 - 1/8 - 1/4.*
In Fig. 11A when the tube had descended to 850 mb, the atmospheric pressure data were
changed to simulate an updraft (lowering outside pressure) followed by a downdraft. The most
recent 7 mass packets were lost from the tube during the updraft, and replaced by new air during
the downdraft (above average rate of increase of outside pressure). As a result, the air sample
that just escaped from being lost is now adjacent to the replacement air, creating the jump in rms
mixing because it has been ~15 s longer in the tube than the first replacement air entering. In Fig.
11B the AirCore has now three sections, from the open to the closed end, first 30.1 m of 1/4",
then 52.1 m of 1/8", and 10.1 m of 1/4" diameter, which we will call "1/4 - 1/8 – 1/4". This was
done solely to illustrate clearly the effects of using different diameters. Similar to what we saw in
Fig. 10A, the spread of mixing steepens near the closed end. Also those samples resided not long
enough in the 1/8" section to have much benefit in terms of slowing down the mixing, but
between 0.80 and 0.85 they had been long enough in the 1/8" section to have experienced less
mixing than air ending up at the 0.57 point, the first transition between 1/4 and 1/8".
We will now express the amount of spreading (in both directions – twice the rms distance) of
each equal-mass "packet" of air as a fraction of the total mass of air in the tube, assuming that
the temperature inside the tube has become uniform. If that fraction were 0.01 everywhere in the
tube there would be slightly less than 100 independent samples in the AirCore. Slightly less
because the remaining fill air in the back takes up space. Figure 12 shows a more realistic
situation. Each sample takes up the same volume, separated by the blue vertical lines, producing
vertical boxes. If there is almost no mixing, as in the case of the last sample that entered the
AirCore, the sample almost completely fills the first volume (or box in Fig. 12A), which is
indicated by the value 1.0 on the y-axis. The red curve centered on the second box has started to



"leak" some sample into the adjoining boxes. The next samples shown are the 7[th], 12[th], and 17[th].
For the latter, the sample is just starting to leak into boxes 15 and 19. To plot the start of this
process correctly, each packet is subdivided into 13 equal portions. Narrow Gaussian spreading,
slowly increasing further into the tube, is calculated for each portion, and then summed. The
width of each Gaussian is shown in figure 10C as a function of fraction of cumulative mass in
the tube, and the area of each curve is 1/13 of the area of the box. This produces a constant value
of 1.0 in the center and only the outer portions reach into the neighboring boxes.

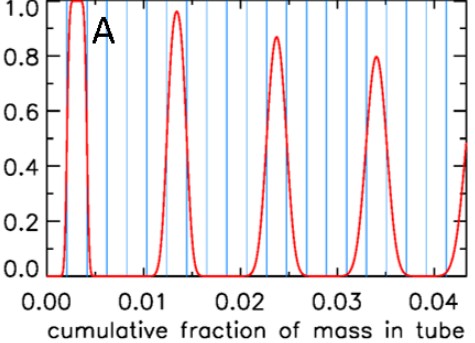 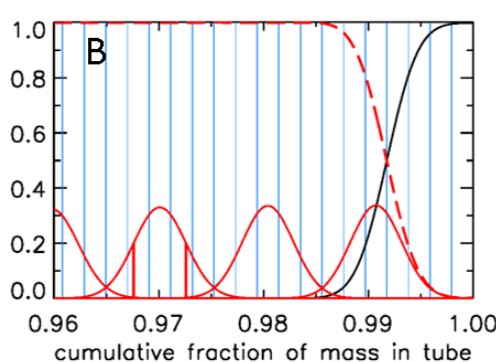


*Figure 12. A, Mixing of individual air "packets" (red) near the open end with their neighbors*
*after valve closure for the case shown in Fig. 10C; B, mixing near the closed end (red), vertical*
*red lines centered on 0.97 show the ± 1 σ points, black curve is remaining fill air, and the sum of*
*all actual sample packets, also of those not shown, is the red dashed line.*
In Fig. 12B we plot the situation near the closed end. As in Fig. 12A, the mixing of only every
fifth air packet is plotted, here ending with the first that came in at the highest altitude, centered
approximately at 0.991. The remaining fill air in this case has the mass of four packets, and the
curves of fill air and of the total air sample (sum of all packets) cross over at exactly the point
where the fourth box from the right starts. How we calculate mixing at a closed end (at x = 0) is
shown in Fig. 13.

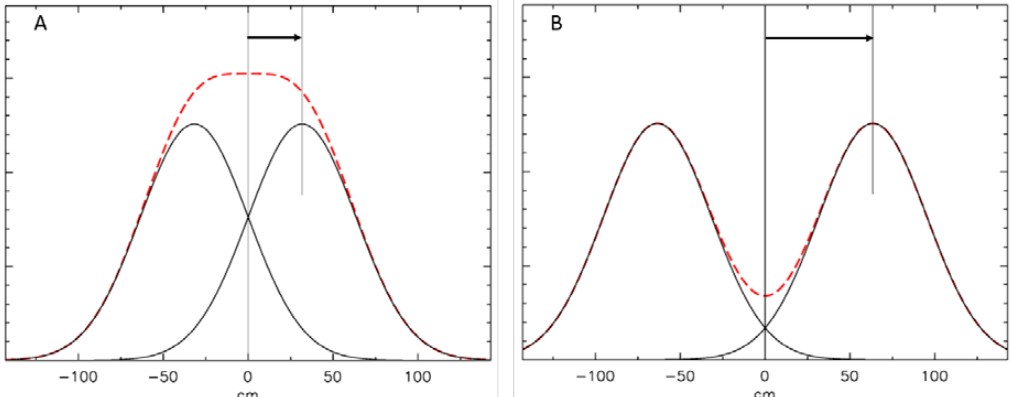


*Figure 13. Mixing at a closed end. The Aircore is to the right of the zero centimeter point. A, the*
*distribution of mixing started one hour ago from a plane at 31.8 cm (one root-mean-square of*
*the distribution), indicated by the arrow. A fictitious "mirror" distribution is centered at -31.8*
*cm. The red dashed curve is the sum of the two distributions; B, same calculation, but the center*
*of the distribution is twice as far from the end as in A.*

Diffusive mixing that would be to the left of x = 0, is reflected toward positive values of x. The
slope of the distribution must be zero at x = 0 because any non-zero slope would imply a
diffusive flux out of, or into, the tube. This is conveniently modeled by assuming a fictitious
distribution mirrored relative to x = 0, then the two are added, and the portion of the sum for
positive values of x represents the mixing distribution near a closed end.

Let us assume that after the valve has been closed there has been a half hour delay before
analysis starts. Therefore, additional diffusion has taken place, as shown in Fig. 14 for the case
1/4 - 1/8 –1/4 (Figure 11B). The 2Dt term has been increased by an amount dependent on the
diameter of the tube, normalized as in Eq. 7. In the upper right (panel B) the square root of the
sum has been taken, and then transformed into the spreading width relative to total mass in the

485

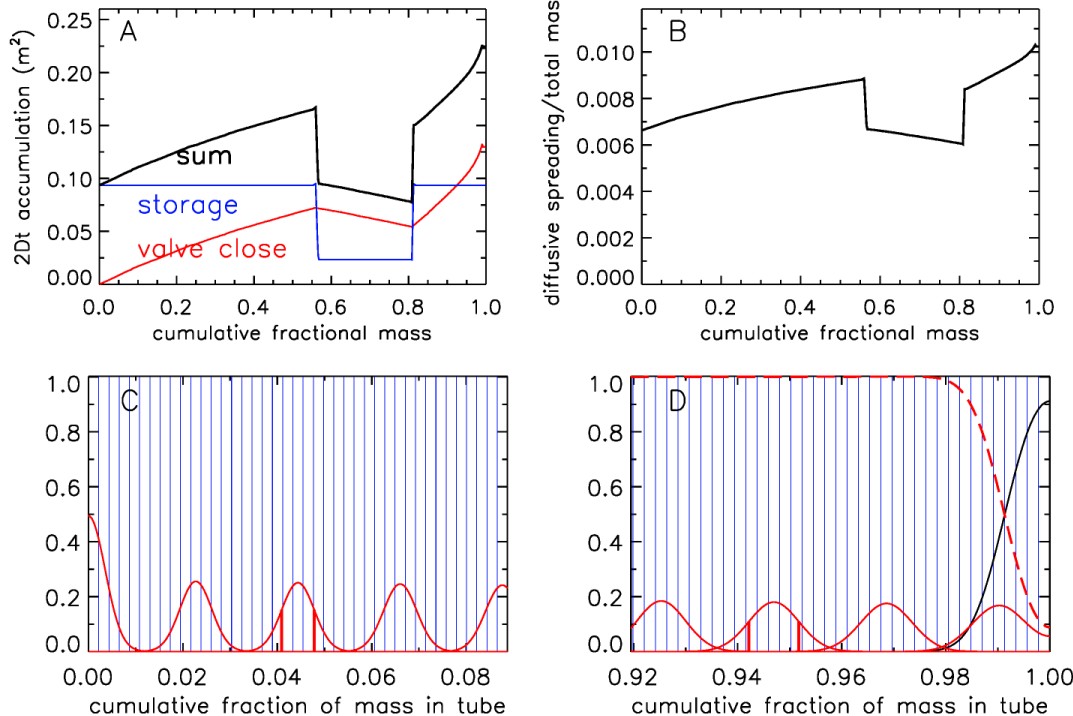

486

*Figure 14. Mixing after 30 minutes of storage, for AirCore 1/4 - 1/8 -1/4. A, Sum (black) of the*
*2Dt accumulation during the flight (red) and during storage (blue), in units of $m^2$; B, spreading*
*width expressed as a fraction of total mass in the tube; C, amount of spreading near what was*
*the open end, for clarity only every $10^{th}$ packet is shown; D, same, near the closed end. Vertical*
*red lines show the $\pm 1 \sigma$ distances from the peak.*

tube. The width is defined here as the distance between the $\pm 1 \sigma$ points of the Gaussian which
contains ~68% of the probability distribution, shown in Fig. 14C at x = 0.0410 and 0.0478
around the center at x = 0.0444, and in fig. 14D at x = 0.9422 and 0.9518 around the center at x =
0.9470. These numbers correspond to the full widths shown in fig. 14B. The last packet to enter
the tube is centered at x = 0.0011 and 1 $\sigma$ = 0.0033. Most of the diffusive spreading is to the
right, so that the peak is almost twice as high and the full width a little over half as wide as the
one centered on x = 0.023.

Often the AirCore is analyzed significantly later than 30 min. after valve closure, and the
measurement process itself may take half an hour. In Figure 15 the state of mixing four hours
after valve closure has been calculated, and two AirCore configurations are compared. The



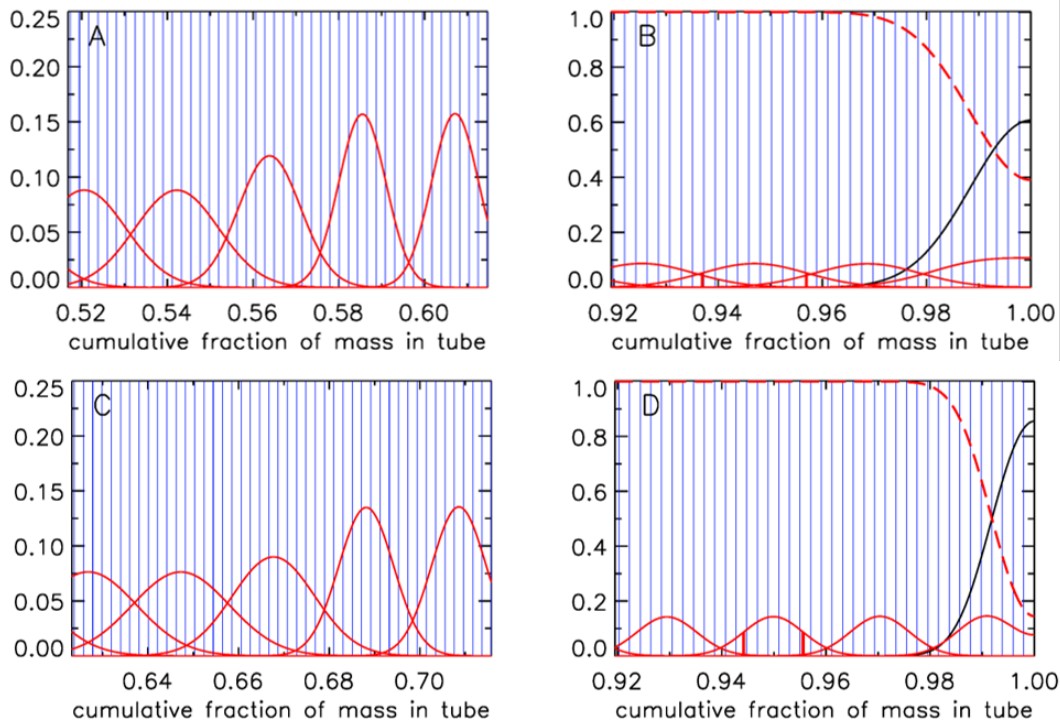

502

*Figure 15. Mixing after 4 hours of storage. A, at the transition from 1/4" diameter to 1/8", for*
*AirCore 1/4 - 1/8 - 1/4; B, near closed end, for 1/4 - 1/8 - 1/4; C, at the transition from 1/4"*
*diam. to 1/8", for AirCore 1/4 - 1/8; D, near closed end, for 1/4 - 1/8.*

spreading width of air "packets" near the closed end is nearly twice as large for the 1/4 - 1/8 –
1/4 case as for the 1/4 - 1/8 case, and the initial fill air penetrates almost 50% further into the
tube. It would in most cases not be a good idea to have a wide bore section at the closed end. If
one waits 24 hours (6 times longer) before starting the analysis, the spreading width near the
closed end, centered at x = 0.9470, is 2.32 times larger than after 4 hours, not quite √6 because
after 4 hours the spreading that occurred during the descent still makes a small, but still
noticeable, contribution.

## 7. Potential information content of the AirCore

When the air is slowly pushed through an analyzer, we obtain a continuous curve for the mole
fraction of the gases of interest as a function of fractional cumulative mass in the tube which is
linked to flight data such as pressure altitude, geometric altitude, latitude, longitude, etc. as
calculated from the filling dynamics. We define the information content as the number of
independent air samples that are inside the tube, or the number of degrees of freedom (DoF).





Longer wait times before analysis decrease DoF, but it could be decreased further by additional
mixing in the measurement cell, or by successive analyzer cells measuring different gas species.
In the section above we chose more than 400 equal mass packets to calculate mixing. This was
done to prevent a possibly low numerical resolution of the mixing calculation which would
unnecessarily create a low bias in DoF estimates. Ideally, the measurement process could be
modeled in a way similar to the fill and mixing calculation above, convolving the packets
leaving the AirCore with a pulse response of the measurement cell. The response could be
measured separately by introducing a sharp spike just before the cell, and recording how it is
mixed and flushed out. This would be similar to the spiking method described by Wagenhäuser
et al. (2021). In the worst case the measurement cell would be perfectly mixed giving rise to
exponential flushing. After one cell volume has entered from the AirCore into the measurement
cell, the latter still contains a fraction 1/e of what went through the cell before, so that the new
volume comprises $(1 - 1/e) = 0.63$ of the cell loading. On the other hand, "plug flow" (like in the
AirCore itself) would produce very little additional mixing, but there could still be some
turbulent eddies near the entrance and exit of the cell. The actual influence of the measurement
cell on mixing lies somewhere in between those two extremes.

## 8. Numerical implementation

The AirCore can consist of one or more sections of different length, each with a different inner
diameter. For example, GML has flown AirCores with a wider bore at the open end and a narrow
bore at the closed end, in order to get better vertical resolution for the stratosphere. The sections
can be divided into a number of smaller segments when Eq. 2 is discretized for numerical
solution (Figure 16):

$$Q = -\rho \frac{\pi r^4}{8\eta} \frac{dP}{dz} \implies Q_j = -\frac{P_j + P_{j+1}}{R(T_j + T_{j+1})} \frac{\pi r_j^4}{8\eta_j} \frac{P_{j+1} - P_j}{dz_j}$$

$Q_j$ is centered in the middle of segment $dz_j$. The first factor in $Q_j$ is the average amount density
$(\rho_j)$. The pressure change at the boundary between segments $dz_{j-1}$ and $dz_j$ caused by the
imbalance of the flows $Q_{j-1}$ and $Q_j$ is equal to that imbalance divided by the volume between the
mid points of $dz_{j-1}$ and $dz_j$. Adding in the pressure change due to temperature (Eq. 2) we get for
the change at boundary j:

$$\frac{dP_j}{dt} = \frac{P_j}{T_j} \frac{dT_j}{dt} + \frac{T_j}{0.5(dz_{j-1}r_{j-1}^2 + dz_j r_j^2)} \frac{P_j + P_{j+1}}{T_j + T_{j+1}} \frac{r_j^4}{8\,\eta_j} \frac{P_{j+1} - P_j}{dz_j}$$

$$- \frac{T_j}{0.5(dz_{j-1}r_{j-1}^2 + dz_j r_j^2)} \frac{P_{j-1} + P_j}{T_{j-1} + T_j} \frac{r_{j-1}^4}{8\,\eta_{j-1}} \frac{P_j - P_{j-1}}{dz_{j-1}} \qquad \text{Eq. 8}$$





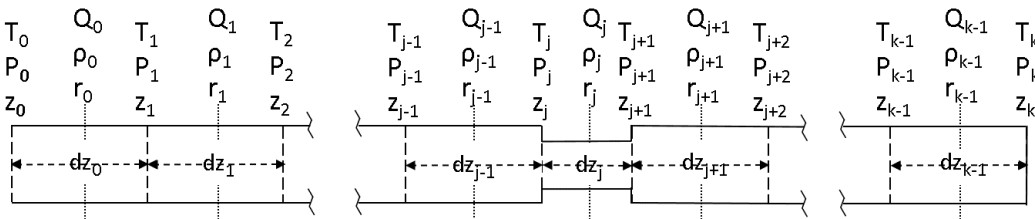


*Figure 16. Coordinate system in the AirCore. The coordinate along the length of the tube is z (m).*
*There are k segments, starting from the open end at $z_0$ to the closed end at $z_k$, between the*
*vertical dashed lines. Amount flow ($Q_n$, mol $s^{-1}$), amount density $\rho_n$ (mol m-3), simply written as*
*Q and $\rho$  from here on out, are defined in the middle of each segment, pressures (P) and*
*temperatures (T) are defined at the borders of each segment. The length (dz) as well as radius*
*(r) of the segments may differ.*
The first term (P/T)(dT/dt) is handled separately from the two other terms describing the amount
change. We write the latter two with the time step going from n to n+1 (superscript):
$$P_j^{n+1} - P_j^n = \left[\frac{2T_j^n(P_{j+1}^n + P_j^n)}{T_{j+1}^n + T_j^n}\frac{r_j^4}{\eta_j}\frac{P_{j+1}^{n+1} - P_j^{n+1}}{dz_j}\right.$$

$$\left. - \frac{2T_j^n(P_{j-1}^n + P_j^n)}{T_{j-1}^n + T_j^n}\frac{r_{j-1}^4}{\eta_{j-1}}\frac{P_j^{n+1} - P_{j-1}^{n+1}}{dz_{j-1}}\right]\frac{t^{n+1} - t^n}{8(dz_{j-1}r_{j-1}^2 + dz_j r_j^2)} \qquad Eq. 9$$

On the right hand side we have defined the pressure *differences* at the *end* of the time step. The
reason is to make the solution of the matrix equation described below unconditionally stable.
This method has been described as "fully implicit" or "backward time" (Press, 1992). We leave
the pressure and temperature *averages* as defined at the start of the time step. They determine the
average amount density of the air and do not create any numerical instability. Eq. 9 can be
further re-arranged, for j =1 to k-1, as
$$P_j^n = -\frac{t^{n+1} - t^n}{8(dz_{j-1}r_{j-1}^2 + dz_j r_j^2)}\left[\frac{2T_j^n(P_{j+1}^n + P_j^n)}{T_{j+1}^n + T_j^n}\frac{r_j^4}{\eta_j dz_j}\right]P_{j+1}^{n+1} +$$

$$\left(1 + \frac{t^{n+1} - t^n}{8(dz_{j-1}r_{j-1}^2 + dz_j r_j^2)}\right)\left[\frac{2T_j^n(P_{j+1}^n + P_j^n)}{T_{j+1}^n + T_j^n}\frac{r_j^4}{\eta_j dz_j} + \frac{2T_j^n(P_{j-1}^n + P_j^n)}{T_{j-1}^n + T_j^n}\frac{r_{j-1}^4}{\eta_{j-1} dz_{j-1}}\right]P_j^{n+1}$$

$$- \frac{t^{n+1} - t^n}{8(dz_{j-1}r_{j-1}^2 + dz_j r_j^2)}\left[\frac{2T_j^n(P_{j-1}^n + P_j^n)}{T_{j-1}^n + T_j^n}\frac{r_{j-1}^4}{\eta_{j-1} dz_{j-1}}\right]P_{j-1}^{n+1} \qquad Eq. 10$$

This is a tridiagonal matrix equation, $\mathbf{A} \bullet P^{n+1} = P^n$ , linking the k+1 dimensional pressure vector
$P^{n+1}$ at the end of the time step to the pressure vector $P^n$ at the start of the time step. The solution



is $P^{n+1} = A^{-1} \cdot P^n$, in which $A^{-1}$ is the inverse matrix calculated by the subroutine TRISOL which is
the IDL version of TRIDAG described by Press et al (1992). If the tube is closed at z = 0, then in
the first line of **A** the first (diagonal) and second (above the diagonal) element (all others are
zero) are respectively
$$1 + \frac{t^{n+1} - t^n}{8(dz_0 r_0{}^2)} \frac{2T_1^n(P_1^n + P_0^n)}{T_1^n + T_0^n} \frac{r_0^4}{\eta_0 dz_0} \quad \text{and} \quad -\frac{t^{n+1} - t^n}{8(dz_0 r_0{}^2)} \frac{2T_1^n(P_1^n + P_0^n)}{T_1^n + T_0^n} \frac{r_0^4}{\eta_0 dz_0}$$

If the tube is open at z = 0, then the first element of the first line equals 1, and all others are zero.
In this case $P_0$ is defined at all times by the outside atmospheric pressure, or by a defined
pressure from a cylinder. There is no influence from any place inside the tube. The algorithm
also allows the other end to be either closed or open to outside air. If closed, then the last two
elements of the (k+1)[st] row are respectively
$$-\frac{t^{n+1} - t^n}{8(dz_{k-1} r_{k-1}^2)} \frac{2T_{k-1}^n(P_{k-1}^n + P_k^n)}{T_{k-1}^n + T_k^n} \frac{r_{k-1}^4}{\eta_{k-1} dz_{k-1}} \quad \text{and}$$

$$1 + \frac{t^{n+1} - t^n}{8(dz_{k-1} r_{k-1}^2)} \frac{2T_{k-1}^n(P_{k-1}^n + P_k^n)}{T_{k-1}^n + T_k^n} \frac{r_{k-1}^4}{\eta_{k-1} dz_{k-1}}$$

If both sides are open, each with a different defined constant pressure, then after an initial
transient the flow settles to steady state flow corresponding to Poiseuille's equation.
This describes the core algorithm, of which there are two versions, called tubeflowstep3.pro and
tubeflowstep3Cv.pro. They have been programmed in Interactive Data Language (IDL). These
algorithms have the flexibility to accommodate segments of the tube that have different lengths
as well as diameters, flows in both directions, one or two valves open, a temperature gradient
along the tube with its corresponding viscosity gradient, and variable time steps. Another
routine, called analyzefill_Gaus_ict.pro, reads the lengths and diameters of tube sections, valves
and dryer, and the relevant flight data, namely outside air pressure and temperature, the
temperature of the AirCore at different locations along the tube, all as a function of time. If $C_v$
and $X_{TPR}$ values of valves are defined they will be used. In that case tubeflowstep3Cv.pro nudges
the apparent internal diameter of one or more valves for a given flow toward satisfying Eq. 5 (see
section 5). This needs to be iterated because when we change the internal valve diameter the
pressures and flows will then adjust elsewhere in the tube. The analyzefill_Gaus_ict program
also reads altitude, latitude, and longitude, but they are not needed for the flow dynamics
calculation per se. It sets up the coordinate system and initializes variables. By calling
tubeflowstep3.pro at every time step, or tubeflowstep3Cv.pro if $C_v$ and $X_{TPR}$ values are defined,
it calculates the pressure in the tube, the amount of air and the amount flow, and the flow
velocity, all as a function of time and location in the tube. This is how altitude, pressure altitude,
latitude, and longitude are tied to position in the tube. The _Gauss portion of the name indicates



that Gaussian mixing is used as described in this paper, and _ict indicates that the program
expects the needed information about the tube and the flight in the ICARTT format.
Although developed simultaneously with analyzefill_Gaus_ict.pro for the passively filled
AirCore, the tubeflowstep3Cv program can also be used to model flow when the AirCore is
actively filled with a pump and some form of flow and pressure control. In that case a program
equivalent to analyzefill_Gaus_ict.pro would need to be developed.
The code in analyzefill_Gaus_ict.pro also produces diagnostic graphics showing how the fill
proceeded. In fact, all figures in this paper have been produced by analyzefill_Gaus_ict.pro
except for Figure 9.

**Acknowledgement**

I thank Anna Karion, Colm Sweeney, Tim Newberger, Jack Higgs, Sonja Wolter, and Bianca
Baier for making our lab's AirCore program blossom. Especially the controlled return is a very
promising improvement over the return by parachute.

The programs are available at ….

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
