# Peer review of "Fill dynamics and sample mixing in the AirCore"

_Atmospheric Measurement Techniques, 2021_

## Author Response (AR1)

Reviewer #1

Comment 1: At this time the dryer is modeled only as a short piece of tubing with a porosity of 0.3, meaning that 30% of the volume is air, 70% is large pieces of magnesium perchlorate. Permeability is a different property that we don't know at this time. The flow resistance of the dryer is implicitly added to the flow resistance of the valve because the valve parameters have been optimized to match the calculated history of the pressure drop across the entire tube to the measured history. I added a new section 9 with recommendations for further improvements. One recommendation is laboratory testing of valves and dryers.

Comment 2: I don't expect the dryer to have much influence on the mixing because the length is short. The turbulence generated by the dryer will die out very quickly as estimated in section 3, the paragraph about inertial effects.

Technical suggestion 1: I added dashed vertical lines to figure 3 to make it easier to read.

Technical suggestion 2: I added a comparison table of the two AirCores, but I prefer to keep each figure close to the place where it is discussed.

Table 1

| AirCore Trainou 2019 | Int. Dia (mm) | Length (m) | | Aircore Oklahoma 2013 | Int. Dia (mm) | Length (m) |
|---|---|---|---|---|---|---|
| Aircore tubing | 2.16 | 0.76 | | AirCore tubing | 5.84 | 25.9 |
| AirCore tubing | 2.92 | 91.5 | | AirCore tubing | 2.67 | 36.6 |
| AirCore tubing | 2.16 | 0.76 | | | | |
| internal volume | | 619 cc | | internal volume | | 890 cc |
| | | | | | | |
| Fill history | Time (s) | Altitude (hPa),(km) | | Fill history | Time (s) | Altitude (hPa), (km) |
| start descent | 0 | 7.7,  32.4 | | start descent | 0 | 4.6,   34.6 |
| start fill | 19 | 8.5,   31.6 | | start fill | 2 | 4.7,   34.4 |
| half fillrate | 123 | 17.4, 27.3 | | half fillrate | 7 | 6.3,    33.2 |
| full fillrate | 266 | 34.2, 23.1 | | full fillrate | 58 | 10.4,  30.2 |

Reviewer #2

I don't know what to make of this anonymous reviewer's comments. The reviewer says that the manuscript provides a crucial piece that is missing in the literature about AirCore, that it addresses relevant scientific questions, presents novel concepts (I know because I worked out all of the ideas in this paper), and that the writing is very readable. I purposely did not provide an overview of the literature in the introduction. I did include information about the genesis of the AirCore idea and early tests and failures because it is unknown to most readers. In response to the reviewer's concern I added a statement (lines 61-63, all line numbers refer to the revised manuscript in MS Word track changes

mode) that refers the reader to two recent overviews of the literature. I also included in the introduction a statement about a fundamental difference between in-situ sampling methods and remote sensing that very few people appear to be aware of, yet is very relevant to AirCore.

With respect to the second major point, about the lack of coauthors, I need to give some background for my decision. I worked on the code base for the AirCore over a long period of time, since 2005. It took a long time because there were many long pauses in between. One could fault me for that. I gave the code to the AirCore team in 2013, including subroutines and programs to test the code. I helped with some problems that were encountered in translating from IDL to Matlab. A few years later it was translated from Matlab into Python. Again I helped out, but I had some misgivings because my IDL code had improved significantly from the 2013 version. One of the improvements was to remove the assumption that the tube had a uniform temperature. My 2017 code allowed for temperature to vary in time and along the tube. At that time we had typically five thermistors along the tube. It is relevant to the flow dynamics because the viscosity of air becomes lower at low temperatures, such as in the stratosphere. By then (2017) I felt that I was gradually being excluded from the AirCore project. Also around 2013, I suggested that we measure differential pressure between the closed end of the tube and outside air because it could provide a crucial diagnostic of the dynamics calculation. In 2014 I did an analysis of "slug tests" going specifically through the analyzer only. That 2014 analysis is the basis of my statement (lines 562-563) that the analyzer response is between well-mixed and plug flow. I shared the results with the AirCore team, but I did not receive any feedback. With respect to the differential pressure measurements, I knew that at least a few flights had them, but I had no idea that it had become pretty much standard practice. In summary, this paper is strictly about the fill process and associated sample mixing, not about how the measurements are done or what they show. The analysis in this paper, and the understanding it affords, has been entirely mine. The same applies to the code, it was a one-way stream. The treatment of valves and mixing in the current version of the code is more recent, it was developed during 2020-21. The contributions of the NOAA AirCore team are noted in the Acknowledgements.

There are many further improvements that can be made. I have added a new section 9 to the end of the paper in lines 647-665, suggesting some avenues of potential work. I remain very willing to cooperate on such work, and co-authorship would follow naturally from that.

With respect to open availability of the code, of course that will happen.

Reply to detailed comments (answers refer to the line numbers in the revised text):

p. 1, 10 (the reviewers comments refer to line numbers in the originally submitted manuscript) I use the word "we" repeatedly because I like a conversational style, as if am looking at the material together with the reader.

p.1, 16 Introduction. This paper is not about the state of the field, it is strictly about how the AirCore works. Currently I have put into the introduction some material about the genesis of the AirCore idea and very early work that we did, and secondly a paragraph about the importance of calibrated measurements, something that not many readers are familiar with in my experience, but it is extremely

important for the value AirCore can bring.  Then, for the reader's convenience I have added some references of recent papers that describe the status of the field (lines 61-63)

p. 1, 26 The expression $X_{rms} = (2Dt)^{0..5}$  is in every text book that treats molecular diffusion. I prefer to leave it here as is.

p. 1, 32 affiliations.  I have done that in lines 32, 41-42, 672-673

p., 32-37  analytical methods     Analytical methods are not the subject of this paper.

p. 2, 59    Done, in line p. 2, 60

p.2, 67  0 degrees C?   The expression for Xrms is valid for any value of D, while D depends on temperature. It is treated in lines 221-228 in the paper.

p.3, 74-75    I thought that capillary effects are well known. In any case, I added line 79.

p. 3, 95-96  The main findings of this paper were not at all influenced by the university students experiments. I was their mentor. Their experiments motivated me to write the first version of the code, as described on lines 94-99. So I changed "we were" to "I was" on line 98. In this case the use of "we" is not good, and I thank the reviewer for catching it.

p. 3, 101-102   There is a dryer right at the entrance of the AirCore, so that using the molecular weight of dry air is appropriate (line 109)

p. 4, 108   "very nearly"   I actually explain in lines 117-118 that the mean free path is inversely proportional to density, so that it cancels out the factor "$\rho$" in the expression $\eta \approx (1/3) \rho \, \mathbf{c} \, \lambda$. I have now explicitly mentioned this on line 119.  Also, please note that the entire section 3 of the paper is about approximate magnitudes.

p. 4, 112-114    I cannot see the benefit of merging these two sentences.

p. 4, 112-113   My answer is above.

p. 4, 122      I made a change on line 129

p. 5, 164 and 167   The publication date of Berg's paper should have been 2005. I have changed it on lines 172 and 175, so that it now corresponds to the reference list. I thank the reviewer for catching this.

p. 8, 236   I have made changes to Figs. 1 to 5, 7, 8 to make them easier to read.

p.8, 238    I don't like "nearly constant near the tropopause"

p.11, 321    I changed the text from "minuscule" to "negligible", which is more precise.

p. 24, 620   This has been fixed on lines 677-678